# Caffeine Protects Against Hyperoxia-Induced Structural Lung Injury and Restores Alveolar Development in Neonatal Rats

**DOI:** 10.3390/antiox14121497

**Published:** 2025-12-12

**Authors:** Stefanie Endesfelder, Christoph Bührer

**Affiliations:** Department of Neonatology, Charité—Universitätsmedizin Berlin, Augustenburger Platz 1, 13353 Berlin, Germany; christoph.buehrer@charite.de

**Keywords:** hyperoxia, oxygen, postnatal immature lung, caffeine, morphology, fibrosis

## Abstract

In the developing lung, oxidative stress caused by relative hyperoxia constitutes a central pathogenic mechanism of neonatal lung injury resulting in bronchopulmonary dysplasia (BPD). The immature postnatal lung is highly susceptible to oxidative damage due to incomplete antioxidant defenses and ongoing alveolar and vascular maturation. In a postnatal high-oxygen-induced rat model of BPD-associated lung injury, three or five days of exposure to 80% oxygen was found to disrupt developmental signaling pathways, downregulating genes essential for alveolarization and angiogenesis while inducing profibrotic mediators and collagen expression (Sirius Red staining). These changes resulted in simplified alveolar architecture, as quantified by toluidine blue staining and mean linear intercept analysis of normalized volumes of parenchyma, non-parenchyma, airspaces, septa, and edema. Acting as a multifunctional antioxidant with antifibrotic activity, caffeine mitigated structural lung damage and normalized the transcription of angiogenic and fibrotic genes. It counteracted TGF-β/CTGF-driven fibrogenic signaling and promoted recovery of normal lung morphology following hyperoxic injury. Under normoxic conditions, however, caffeine transiently upregulated profibrotic mediators. Overall, caffeine mitigates hyperoxia-induced lung injury and may actively promote physiological lung maturation, warranting future studies to define optimal dosing windows, clarify context-dependent fibrotic signaling, and translate gene-level effects into long-term clinical outcomes.

## 1. Introduction

The prevention of bronchopulmonary dysplasia (BPD) and the reduction in respiratory instability remain major challenges in neonatal intensive care. Improved survival rates are accompanied by a higher rate of chronic respiratory complications [1,2]. Extremely and very preterm infants (<28 weeks of gestational age) are especially susceptible to respiratory instability evolving into (BPD) [3]. Despite advances in neonatal intensive care medicine, such as the increased use of non-invasive ventilation and optimized surfactant therapy, the incidence of BPD remains high or is even increasing, while mortality is declining [4,5].

BPD is one of the most common respiratory complications in premature infants. Similarly, preterm infants without BPD, as well as moderate and late preterm infants, may also exhibit relevant and sometimes persistent impairments in lung function and an increased risk of respiratory morbidity [6,7,8]. Lifelong morbidities affecting various aspects of quality of life can result from respiratory diseases following premature birth. Cohort studies show that preterm infants with very low gestational age (<32 weeks) have a significantly increased risk of persistent and progressive lung function deficits, asthma, and the development of chronic obstructive pulmonary disease in adulthood [9,10,11]. These morbidities are reflected in higher hospitalization rates, greater medication requirements, and an increased prevalence of respiratory symptoms and infections already in childhood, often with functional limitations persisting into adulthood [12].

Current intervention options for the treatment of BPD primarily involve a combination of non-invasive ventilation, targeted pharmacotherapy, and supportive measures. The most important evidence-based therapies include the early application of non-invasive ventilation immediately after birth, volume-controlled ventilation, and the minimization of oxygen toxicity to prevent invasive ventilation and the associated lung injury [13,14,15], as well as surfactant therapy, preferably using less invasive surfactant administration [16]. Corticosteroids are employed to reduce the incidence of BPD; however, the optimal timing, choice of agent, and dosage remain debated [17,18]. Vitamin A supplementation may lower the risk of BPD but is not routinely used due to uncertain cost–benefit considerations. Diuretics and bronchodilators are applied symptomatically in established BPD, improving lung function in the short term but showing no preventive efficacy [14]. Experimental approaches include clinical and preclinical studies investigating mesenchymal stromal cells (MSCs), IGF-1/IGFBP-3, and IL-1 receptor antagonists [13,19,20]. Addionally, new therapeutic concepts, such as systems pharmacology therapeutics, aim to interrupt the complex, self-reinforcing cycle of BPD progression, comprising oxidative stress, inflammation, and downstream endoplasmic reticulum (ER) stress, to promote proper lung regeneration rather than defective repair [21]. While preclinical data indicate improved lung development and reduced inflammation, clinical benefits remain to be confirmed. The prophylactic and therapeutic use of caffeine is well established and significantly reduces the risk of BPD [14,15,22].

BPD is characterized by a complex interplay of impaired lung development and dysregulated processes of injury and repair [23]. Morphologically, this is manifested by a simplified alveolar architecture with reduced septation and enlarged alveoli, resulting in a diminished surface area for gas exchange [24,25,26]. In parallel, pulmonary microvascular development is impaired, leading to dysmorphic angiogenesis with reduced capillary density and an increased risk of pulmonary hypertension [24,25,27]. The small airways also exhibit pathological alterations, including wall thickening, smooth muscle hyperplasia, and enhanced reactivity, which contribute to obstructive ventilatory disorders and increased susceptibility to infection [24,27]. Interstitial remodeling processes involving fibrosis and chronic inflammation further compromise lung elasticity and compliance [25,26].

Ventilation-induced damage aggravates these changes by causing endothelial and epithelial dysfunction and disrupting the integrity of alveolar-capillary barrier. This results in pulmonary edema, increased protein permeability and diffuse alveolar damage, which progresses through exudative, proliferative, and fibrotic phases characterized by distinct patterns of tissue remodeling. In premature infants, the lungs are particularly vulnerable as the structural and functional maturation of both endothelium and epithelium is incomplete [28,29]. Even short periods of mechanical ventilation can lead to significant parenchymal and bronchiolar injury. Although repair mechanisms are activated, the risk of developing chronic lung disease such as BPD remains high [30].

Oxidative stress resulting from hyperoxia and the formation of reactive oxygen species (ROS) is central to BPD pathogenesis [31,32]. Preterm infants are particularly susceptible due to immature antioxidant defense systems [33,34]. Reduced glutathione concentrations limit free radical detoxification, promote inflammation and apoptosis, and disrupt alveolar and vascular development [35,36]. Elevated proinflammatory cytokines such as IL-6, IL-8, and TNF-α contribute to chronic tissue damage and impaired alveolarization [37,38], with an increased neutrophil-to-lymphocyte ratio indicating systemic inflammation [38]. Additional mechanisms include impaired VEGF signaling and angiogenesis [39], epigenetic modifications [38], and programmed cell death pathways such as apoptosis, autophagy, and ferroptosis [32]. Genetic predispositions and dysregulated immune responses, including Th1/Th2 imbalance, further increase risk [23,38]. Overall, oxidative injury constitutes the major molecular risk landscape BPD progression.

Given the central role in premature lung injury, antioxidant strategies, including potential effects of caffeine, may hold therapeutic relevance. The interaction between oxidative stress and inflammation, although critical, remains incompletely understood in BPD an is an important area of ongoing [40].

Animal models, particularly mouse and rat models, are essential for investigating these mechanisms under controlled conditions [41]. They allow targeted analysis of oxygen toxicity, antioxidant system dysregulation, and BPD development [42]. The lungs of rodents in the late fetal and early neonatal periods are comparable to human preterm lungs in terms of development and antioxidant capacity [14,42]. Similarities include immature surfactant production, reduced antioxidant enzyme activity, and heightened vulnerability to oxidative stress after birth and oxygen exposure [42,43]. Changes observed in animal models after hyperoxia or intermittent hypoxia, such as alveolar hypoplasia, inflammation, and impaired vascular development, mirror those seen in human preterm infants [42,43,44]. Despite these similarities, differences in lung development, genetics, and immune response limit clinical translatability, so findings must be critically evaluated [42,43,45].

Against this background, animal models are particularly valuable for investigating antioxidant therapies such as caffeine. In mouse and rat models, caffeine exhibits pronounced antioxidant and anti-inflammatory effects. It reduces oxidative DNA damage, modulates antioxidant enzyme expression, inhibits NLRP3 inflammasome and NF-κB activation, and decreases alveolar epithelial cell apoptosis [46,47,48,49,50,51,52]. Additionally, caffeine promotes angiogenesis and alveolar development [53,54]. These effects are dose-dependent, as excessive caffeine concentrations may induce cellular damage [55]. The protective effects observed in animal studies correspond to clinical findings of reduced BPD incidence in preterm infants.

This study aims to investigate the effects of neonatal hyperoxia on postnatal lung development and to explore whether caffeine can modulate these outcomes. We focus on structural, molecular, and remodeling changes, including alveolar simplification, angiogenesis, and expression of key developmental and pro-fibrotic markers. Specifically, we are investigating the acute phase of oxygen exposure within the saccular stage and the transition from the saccular to the alveolar stage of lung development, as well as the subsequent effects in adolescent rats during the alveolar stage without further insults, in order to assess both immediate and longer-term impacts on lung maturation. Furthermore, we aim to determine whether caffeine exerts protective effects under oxidative stress and whether its impact is context-dependent, potentially influencing lung development differently under normoxic versus hyperoxic conditions. The findings of this study are expected to clarify caffeine’s role as a potential modulator of postnatal lung development and its therapeutic relevance for preterm infants at risk of BPD.

## 2. Materials and Methods

### 2.1. Animal Welfare

After an acclimatization period, the pregnant *Wistar* rats were housed individually for two days before labor under controlled environmental conditions with a constant 12 h/12 h light/dark cycle, room temperature, and 60% relative humidity and had access to food and water at all times. In a 12 h birth window, pups were randomized by sex and number and assigned to lactating dams according to the experimental group. The animal experimental procedures were evaluated and approved by the local animal welfare authority (LAGeSo, authorization number G-0088/16), including a prior case number determination. All procedures performed on the animal, as well as the husbandry conditions, were in accordance with institutional and ARRIVE guidelines. For a comprehensive evaluation of the data collected, reference is made to the previously published studies with identical experimental cohorts [50,51,56,57], which have focused on pulmonary and neurodevelopmental analyses.

### 2.2. Experimental Design and Caffeine Administration

As described previously [50,51,56,57], all pups born within a 12 h time window were randomized by sex across litters and assigned to lactating dams according to litter size. Sample size was calculated using G∗Power V3.1.2 [58] for local authority approval. Dams with assigned rat pups were randomly divided into the different test groups. Exposure to two oxygen concentrations, either room air (normoxia, NO) or an 80% oxygen enriched atmosphere (hyperoxia, HY), was performed. Normoxic conditions were maintained in the animal housing, while hyperoxia-exposed pups remained with their dams in an 80% oxygen environment in an incubator (OxyCycler BioSpherix, Lacona, NY, USA). The duration of the modified oxygen environment was applied for two distinct periods: from postnatal day (P)0 to P3 (*n* = 6–8) or from P0 to P5 (*n* = 6–8) for both normoxia and hyperoxia conditions. Compared to neonatal pups, adult rats have limited tolerance to high oxygen concentrations and may stop feeding the pups. To prevent oxygen toxicity in the dams, they were alternated between hyperoxic and normoxic litters every 24 h. Pups were randomized into experimental groups on the day of birth (P0). In addition to the oxygen concentration, the treatment of the pups varied in the application of either caffeine or vehicle. Caffeine or vehicle was administered intraperitoneally (i.p.), correlated to body weight (100 μL/10 g). Crucially, the first application was administered on P0 prior to the onset of the respective oxygen exposure. The subsequent dosing schedule depended on the exposure duration: the P0-P3 subset received a total of two applications (on P0 and P2), while the P0–P5 subset received a total of three applications (on P0, P2, and P4). Validation of plasma caffeine levels confirmed that the dosing regimen was sufficient to achieve concentrations in the pups (8.8–11.2 µg/mL, [50]) that fell within the clinical therapeutic range (5.5–23.7 µg/mL, [59]). Importantly, experimental groups maintained until P15 received no further applications of caffeine or vehicle after the initial exposure period (P3 or P5), thus remaining untreated during the subsequent recovery phase under room air/normoxia. The dose of 10 mg/kg pure caffeine is equivalent to 20 mg/kg caffeine citrate, which is commonly used clinically [60]. This protocol resulted in four distinct groups at each exposure duration: (a) normoxia (NO, control group): application of 21% oxygen with vehicle (phosphate-buffered saline, PBS, Thermo Fisher Scientific Inc., Waltham, MA, USA), (b) normoxia with caffeine (NOC): 21% oxygen with caffeine (10 mg/kg, Sigma-Aldrich, Steinheim, Germany), (c) hyperoxia (HY): 80% oxygen with vehicle (PBS) and (d) hyperoxia with caffeine (HYC): 80% oxygen with caffeine (10 mg/kg). The study relevant examinations were performed immediately after the end of the oxygen exposure (P3, P5) or after recovery in room air at P15 (P3_P15, P5_P15). The experimental design with the timeline, the experimental groups and the outcome parameters is shown below in Figure 1.

### 2.3. Sampling and Processing Lung Tissue

#### 2.3.1. Tissue for RNA Extraction and Paraffin Embedding

At the time of targeted analyses (P3, P5, P3_P15 and P5_P15), as previously described [50,51], pups received i.p. injection of anesthetics (ketamine (100 mg/kg), xylazine (20 mg/kg), and acepromazine (3 mg/kg)). After opening the thorax, the pups were perfused transcardially as described previously. Cold PBS (pH 7.4) was used to perfuse the lung tissue for further transcript analysis. To prepare the whole lung for paraffin embedding, the PBS perfusion was followed by a further perfusion step with cold paraformaldehyde (PFA, 4% (*w*/*v*), pH 7.4, Merck KGaA, Darmstadt, Germany). After exsanguination of the deeply anesthetized rat pups, the lungs were removed and either snap-frozen in liquid nitrogen for gene expression studies and stored at −80 °C until further use, or dehydrated in ethanol for embedding after one day of post-fixation in PFA at 4 °C and then embedded in paraffin.

#### 2.3.2. Tissue for Glycol Methacrylate Embedding

Sedation and deep anesthesia were achieved with an i.p. injection of ketamine, xylazine and acepromazine (see below 2.3.1). The abdomen and thorax were opened and the trachea was dissected free. A tube was inserted into the trachea via the oral cavity and fixed with a ligature in order to start transcranial perfusion. This was performed under constant pressure, first with cold isotonic Ringer’s solution mixed with procaine and heparin, and then with cold glutaraldehyde-formaldehyde solution (4% (*w*/*v*) PFA and 0.1% (*v*/*v*) glutaraldehyde (GA, Carl Roth GmbH, Karlsruhe, Germany) in 0.2 M HEPES, Merck KGaA). Meanwhile, a recruitment maneuver (10 cm H_2_O to 30 cm H_2_O) was performed twice via a connected ventilation system (MidiVent Ventilator Model 849, Harvard Apparatus GmbH, March-Hugstetten, Germany) to optimally inflate the lungs. The tube was then removed and the heart-lung package carefully removed and post-fixed in the perfusion solution (PFA/GA/HEPES) for a further 24 h. Until further use and after removal of the heart, the lungs were stored at 4 °C in 0.15 M HEPES. The volume of the whole lungs was determined according to Archimedes’ principle [61].

Plastic embedding of the left and the right lung lobes in glycol methacrylate (GMA, Technovit 7100; Heraeus Kulzer, Wehrheim, Germany) was performed over three consecutive days. For the embedding procedure, the lungs were washed twice for 10 min each in 0.15 M HEPES (pH 7.4) and then four times for 5 min each with 0.1 M sodium cacodylate buffer. The samples then remained in distilled water for about one hour. Dehydration was carried out using an ascending acetone dilution series with incubation times of 20 min twice each (70%, 80%, 90% and 100%), with the last step being repeated three times. A mixture of the base solution (Technovit 7100) with 100% acetone was added to the tissue for preinfiltration and incubated overnight at 4 °C. On the second day, the preinfiltration medium was replaced by freshly prepared infiltration medium (100 mL Technovit 7100 with 1 g hardener I). The samples were stored overnight at room temperature and shaken gently. Plastic embedding was carried out on the third day. Infiltration medium was mixed 15:1 with Hardener II, mixed well and placed in embedding cassettes for standardized positioning of the lungs. For optimal polymerization, the samples were left at room temperature for at least 24 h. For later sample preparation for cutting, they were glued to plexiglass blocks with superglue and stored cool. Using a rotary microtome, the glycol methacrylate-embedded lungs were cut into 1.5 µm thick sections and mounted on uncoated slides.

### 2.4. RNA Extraction and qPCR

The procurement of lung tissue for the transcript analyses with underlying RNA isolation and subsequent transcription into cDNA has already been described in the work carried out with the same animal cohorts [50,51]. Briefly described again, total RNA was isolated from frozen tissue by acid phenol/chloroform extraction (peqGOLD RNAPure™, PEQLAB Biotechnologie, Erlangen, Germany). The isolated and DNase-treated RNA (2 µg) was reverse-transcribed, then amplified and quantified in a real-time qPCR with the target gene sequences summarized in Appendix A, with detailed abbreviations for the amplified genes. Amplification of target genes was performed using the mastermix qPCR BIO Mix Hi-ROX (NIPPON Genetics Europe, Düren, Germany) in the QuantStudio™ 3 Real-Time PCR System (Applied Biosystems, Carlsbad, CA, USA, Thermo Fisher Scientific Inc.). Using standardized HPRT as an internal reference, expression analyses of selected target genes were performed using the 2^−ΔΔCT^ method [62].

### 2.5. Sirius Red and Toluidine Blue Staining

The paraffin-embedded lungs were deparaffinized and then hydrated with descending alcohol concentrations (EtOH; 96%, 80%, 70%). After the sections were rinsed in distilled water, they were incubated with a saturated aqueous picric acid solution (Sigma-Aldrich) for at least 1 h. The slides were then washed twice with freshly prepared acidified water (0.5% (*v*/*v*) glacial acetic acid in ddH_2_O) and hydrated again with an ascending concentration of alcohol (EtOH; 70%, 80%, 96%) and cleared with xylene substitute (ROTI^®^Histol, Carl Roth GmbH) before mounted on the slide with aqueous mounting medium.

The GMA-embedded lung sections from the young rats were incubated for 15 min at room temperature with an aqueous toluidine blue solution (1% (*w*/*v*) toluidine blue O (Carl Roth GmbH), 2.5% (*w*/*v*) sodium bicarbonate). The sections were then differentiated in deionized water and mounted in a resin-containing medium.

### 2.6. Aspects of Morphologic Analysis

The left toluidine blue-stained lung lobes of all experimental animals were used for quantitative analysis of morphological characteristics. The stereological analyses were performed according to the recommendations of the American Thoracic Society/European Respiratory Society [63] for the quantitative assessment of lung structure, whereby all lungs were embedded in the same orientation and serially cut into 1.5 µm thick sections. From the beginning of the tissue structure in the GMA block, the sections were numbered and selected for further analyses in comparable areas. For analysis, three left lung lobe sections from each animal were imaged at 10× and 20× magnification with the BZX800 microscope (Keyence, Osaka, Japan) using light microscopy with stitching and Z-stack. The images were taken under identical conditions and then analyzed blindly.

The mean linear intercept (MLI) of lung tissue sections was quantified using ImageJ (version 1.53, National Institutes of Health, Bethesda, MD, USA). Toluidine blue-stained images were acquired at 20× magnification, saved in TIFF format, and standardized to identical resolution. Image segmentation was performed by applying a uniform thresholding method across all samples [64]. To obtain chord lengths, horizontal and vertical test line grids were superimposed on the images. For each animal, ten fields per orientation were analyzed, corresponding to a defined test line length. Image processing was automated through a dedicated macro executed in batch mode, which included background correction, thresholding, chord isolation, and export of measurement data. The results were saved in Excel-compatible format. The MLI was calculated as the sum of the lengths [µm] of all counting lines divided by the total number of intercepts with alveolar septa. The mean intercept value per sample was obtained by averaging the values of all sections from the same animal. These means were then used to compare the intercepts of treated animals with those of control animals.

For each left lung examined in this study, a set of predefined morphological parameters was assessed, including the volume of parenchyma (V_V_ (par, lung), [mm^3^]) and non-parenchyma (V_V_ (non-par, lung) [mm^3^]), the volume of alveolar airspace (V_V_ (alv, lung), [mm^3^]), the volume of alveolar septa (V_V_ (sept, lung), [mm^3^]), the absolute surface area of the alveolar septum (S (sept, lung), [cm^2^]), the mean thickness of the alveolar septum (τ (sept), [µm]), and the volume of alveolar edema (V_V_ (edema, lung), [mm^3^]). As an initial step, the total lung volume was determined, since this value served as the essential reference volume against which all subsequent stereological volume estimations were normalized [65]. Histological sections were stained with toluidine blue and imaged at 10× magnification. Images were acquired in JPEG format and standardized to an identical resolution to ensure comparability across samples. The quantification of volume densities was performed using the point-counting method, a well-established stereological technique that allows unbiased estimation of three-dimensional structures from two-dimensional sections. For this purpose, the software STEPanizer (version 1) was employed, which facilitates stereological evaluation by superimposing grids of test points and lines on digital images [66]. The number of points hitting the respective tissue components was recorded, and volume densities were calculated relative to the reference lung volume. From these volume densities, absolute compartment volumes were derived. For each animal, the estimations were based on a systematic sampling of ten images, ensuring a representative coverage of the lung tissue. The measurements were blinded. All values determined for each sample were first averaged across all sections obtained from the same animal. These animal-level mean values were then used for group-level analyses by calculating the average of all animals within the respective experimental groups. Comparisons between treated and control groups were subsequently based on these group means. All stereological results were exported in an Excel-compatible format to enable structured data storage and facilitate subsequent statistical analysis.

Quantification of the red-stained collagen of paraffin-embedded rat lungs was performed by Sirius Red staining using ImageJ (version 1.53, [67]). For the fibrosis analyses, two complete left and two complete right lung lobes were used for each animal, recorded using a 10× magnification with stitching and Z-stack in the light field (BZX800 microscope). To determine the fibrosis per area, a calibration scale was first created for all subsequent analyses. For this purpose, an object of known size was measured and stored in ImageJ as a reference. The threshold value of the Sirius Red-stained areas was determined via ‘colour convolution2’ and the data were collected in a standardized manner. The percentage of the fibrotic area in the image section and the area of the corresponding lung were recorded, so that final fibrotic areas [%] per area [mm^2^] were included in the evaluation. The measurements were blinded. The mean values per sample were calculated by averaging the values of all sections from the same animal and used to compare the cell counts of the treated animals with those of the control animals. The mean fibrosis percentages of the control animals were used as 100% values, as indicated in the figure legends.

### 2.7. Statistic

Box and whisker plots were created for the statistical evaluation, identical to the previous analyses of these cohorts. [50,51,56,57] The interquartile range (box) was shown, with the line representing the median, while the whiskers show the data variability outside the upper and lower quartiles. As already described, the groups were compared using a one-way analysis of variance (ANOVA). In the case of a partially non-Gaussian distribution, the analysis was carried out using the Kruskal–Wallis test or, assuming that the groups do not have the same variances, using the Brown–Forsythe test. Post hoc tests were matched to the respective omnibus test with Bonferroni for ANOVA, Dunn’s for Kruskal–Wallis, and Dunnett’s T3 for the Brown–Forsythe test. A *p*-value of <0.05 was considered significant. All graphs and statistical analyses were performed using GraphPad Prism 8.0 software (GraphPad Software, La Jolla, CA, USA).

## 3. Results

### 3.1. Survival, Growth and Lung Volume

All rat pups survived exposure to high oxygen concentrations. The nursing dams were rotated between hyperoxia- and normoxia-exposed litters, as described, in order to avoid a possible undersupply of the pups. The weight at the time of analysis (P3, P5, and after survival to P15) increased over time according to postnatal development and significant differences between the oxygen and/or caffeine treated pups could not be detected within the different time points (Figure 2a). However, when looking at the percentage difference in weight gain from the day of birth in a randomized distribution of pups, there was a delayed weight gain in caffeine-treated pups with caffeine application under room air conditions at P3 and P5 (Figure 2b). Caffeine-treated rats that remained under 80% oxygen for the first five days of life also showed delayed weight gain compared to both the control and hyperoxia groups.

The lung volumes of the respective rat pups were determined using the fluid displacement method and are equivalent to the lung volume. The lung volumes increase over time due to development. A significant increase in lung volume was observed for the rats exposed to 80% oxygen for 3 and 5 days after recovery in room air (P3_P15, P5_P15) (Figure 2c). The lung volume stabilized at a comparable level to the control animals when rats were treated with caffeine during hyperoxia. This effect was confirmed when lung volume was related to body weight (Figure 2d).

### 3.2. Lung Morphology

#### 3.2.1. Lung Development

Rat lung maturation occurs postnatally, with alveolarization taking place during the first two weeks of life [68,69]. Postnatal exposure of newborn rats (P0) to 80% oxygen (HY) for 3 (P3) or 5 days (P5) resulted in marked alterations in alveolar development (Figure 3). Compared to normoxic controls (NO), hyperoxia caused a simplified alveolar architecture with fewer alveoli and enlarged distal airspaces (HY at P3 and P5). These abnormalities persisted after recovery in room air until day 15 (HY at P3_P15, P5_P15), while caffeine treatment (HYC) mitigated these effects.

Quantitative assessment using the MLI (Figure 4a) revealed that hyperoxia increased MLI values at all time points (P3, P5, P3_P15, P5_P15), reflecting alveolar wall destruction and enlarged airspaces. Caffeine prevented this oxygen-induced increase at P3, P5, and after recovery from short-term exposure, although after 5 days of hyperoxia followed by recovery, MLI values remained slightly elevated. Under normoxia, caffeine had no structural effect.

Stereological analyses were performed to evaluate parenchymal and non-parenchymal lung compartments. Volume and surface densities are relative parameters influenced by changes in the reference space, all volume densities were related to total lung volume. The volume fraction of parenchyma (Vv (par, lung), Figure 4b), comprising alveoli, septa, and distal airways, was similar among groups at P3 and P5 but increased in hyperoxia-exposed pups after recovery (P3_P15, P5_P15). This increase was prevented by caffeine. Non-parenchymal volumes (Vv (non-par, lung)), including conducting airways and large vessels, remained unchanged (Figure 4c).

The fraction of ventilated alveoli and alveolar ducts (Vv (alv, lung), Figure 4d) was unaffected by hyperoxia or caffeine at P3. After recovery, alveolar volume was significantly increased following both 3- and 5-day hyperoxia, an effect blocked by caffeine. Notably, caffeine under normoxia slightly increased alveolar volume at P5. Conversely, the volume fraction of alveolar septa (Vv (sept, lung)) decreased under hyperoxia, consistent with alveolar simplification. Caffeine restored septal volumes to near-control levels except at P3 (Figure 4e). The absolute surface area of the alveolar septum (S (sept, lung), Figure 4f) was markedly reduced by hyperoxia, with a near 50% decrease already evident at P3. At P5, hyperoxia induced only a trend toward reduction, yet both 3- and 5-day exposures resulted in a persistent decrease that remained detectable through P15. Caffeine did not modify these hyperoxia-induced reductions at any time point (Figure 4f). In contrast, the mean thickness of the alveolar septum (τ (sept), Figure 4g) was consistently diminished under hyperoxic conditions at all acute time points (P3, P5), and this reduction persisted after the recovery phase (P15). Caffeine mitigated septal thinning induced by hyperoxia and fully normalized septal thickness by P15. Under normoxic conditions, caffeine alone slightly decreased septal thickness at P5 (Figure 4g).

Exposure to high oxygen concentrations may have increased the permeability of the alveolo-capillary barrier, promoting interstitial fluid accumulation. Hyperoxia increased alveolo-capillary permeability, leading to mild interstitial edema ((Vv (edema, lung) with 4.7% at P3; 3.2% at P5). Caffeine successfully reduced the edema to approximately 1% of total lung volume (Figure 4h), rendering this effect unmeasurable in the overall assessment of lung volume. However, separate assessment at the P15 time point, representing early hyperoxia followed by recovery in room air, showed an increased lung volume, both in absolute terms and relative to the pups’ body weight (Figure 2c).

#### 3.2.2. Gene Expression of Alveolarization- and Angiogenesis-Associated Mediators

The effect of hyperoxia on the developing lung in relation to alveolarization and angiogenesis was investigated by gene expression analysis of pulmonary transcripts.

CycD2 expression was reduced by 3-day and 5-day hyperoxia. This impairment in transcription persisted after recovery under normoxia (Figure 5a). Without persistent downregulation of gene expression, *FGF10*, *PDGFα*, *PDGFRα*, and *PDGFRβ* showed reduced transcription after acute hyperoxia at P3 and P5 (Figure 5c,e–g). *TGFβ* was down-regulated after neonatal hyperoxic insult only after recovery without high oxygen (P3_P15, P5_P15) (Figure 5h) after it had undergone a strong induction of transcription under acute hyperoxia (P3, P5). Excessive induction of gene expression after significant inhibition at P3 and P5 was seen at P15 for *PDGFRα* and *PDGFRβ* (Figure 5f). An increased transcription level could be detected for *FGF2* and Grem1 after a recovery period in room air (P15) without prior hyperoxia-induced downregulation (Figure 5b,d). Caffeine affected transcription of all target genes when administered concomitantly with 80% oxygen exposure and, with few exceptions, caused significant counter-regulation (*CycD2* at P3, P5 and P3_P15; *FGF2* at P3_P15 and P5_P15; *FGF10* at P3 and P5; *Grem1* at P3_P15 and P5_P15; *PDGFα* at P3; *PDGFRα* at P3 and P5; *PDGFRβ* at P3; *TGFβ* at P3, P5 and P5_P15; Figure 5a–h).

When caffeine alone was applied under normoxia, various alveolarization-related factors were modulated. *FGF10*, *PDGFα* and *PDGFRα* were up-regulated at P5 compared to untreated controls (Figure 5c,e–f), with *PDGFRα* expression remaining elevated even after recovery in room air (Figure 5f). Caffeine had an inhibitory effect on the transcription of *PDGFα*, *PDGFRα* and *PDGFRβ* after treatment up to P3 (Figure 5e–g), while *TGFβ* was induced by caffeine at P3 (Figure 5h).

Exposure to 80% oxygen for three days decreased the expression of *Angpt1* (P3, Figure 6a) and was shown to be overexpressed after recovery below 21% (P3_P15). *Angpt2*, on the other hand, was strictly upregulated both after three (P3) and after 5 days of hyperoxia (P5, Figure 6b). P5_P15 then showed a drastic downregulation of *Angpt2* (Figure 6b). As an associated receptor to *Angpt1* and *Angpt2*, Tie2 was decreased under 80% oxygen and then increased at P5_P15 (Figure 6c). *VEGF* and *VEGFR2* were also inhibited in transcription under hyperoxia at P3 and P5, while VEGFR1 was induced at P3_P15 and P5 (Figure 6e). Caffeine affected transcription of all target genes when administered in parallel with 80% oxygen exposure and, with few exceptions, caused significant counter-regulation (*Angpt1* at P3 and P3_P15; *Angpt2* at P3, P5 and P5_P15; *Tie2* at P3, P5 and P5_P15; *VEGF* at P3, P5 and P5_P15; *VEGFR1* at P5; *VEGFR2* at P3 and P5_P15; Figure 6a–f).

When caffeine was administered under normoxia, the transcription of *Angpt1* at P3 and *VEGFR2* at P5 was induced (Figure 6a,f), and an inhibition of transcription by caffeine was detected for *Tie2* at P3_P15 (Figure 6c), *VEGF* at P3 (Figure 6d), for *VEGFR1* at P3 and for *VEGFR2* at P3 (Figure 6e,f).

### 3.3. Fibrosis

#### 3.3.1. Collagen Staining

Histologic analysis of lung sections from normoxia (NO)- and hyperoxia (HY)-exposed rats showed no subjective difference between the groups in the percentage of area covered by collagen after 3-day (P3, top left) or 5-day (P5, top right) exposure (Figure 7), as well as in comparison to caffeine-treated pups under hyperoxia (HYC). However, the caffeine-treated rats (NOC) under room air showed intensified Sirius Red staining over the entire area. However, survival in room air up to P15 showed that 80% oxygen (HY) resulted in increased collagen production over the entire tissue area after both 3 (P3_P15, bottom left) and 5 days (P5_P15, bottom right). Caffeine treatment (HYC) attenuated this hyperoxia-induced effect, and was no longer elevated for the caffeine-treated group under 21% oxygen (NOC). After quantification of the collagen content over the area, these observations were confirmed for all time points and experimental groups (Figure 8).

#### 3.3.2. Gene Expression of Fibroblast-Associated Mediators

The effect of hyperoxia on the developing lung in relation to fibroblast-associated factors was investigated using gene expression analysis of pulmonary transcripts.

This indicated that some transcripts were reduced in expression to P3 and P5 by acute hyperoxia, such as *Acta2* (Figure 9a), *Col1a1* (Figure 9b), *Col3a1* (Figure 9c), *Csf2* (Figure 9d), and *Pu.1* (Figure 9i). Three days of hyperoxia reduced the transcription of *Fn1* (Figure 9h), whereas only 5 days of exposure to 80% oxygen impaired the transcription of *Timp1* and *Timp2* (Figure 9j,k). In some cases, down-regulated transcription persisted after recovery until postnatal day 15 (*Csf2* at P5_P15; *Pu.1* at P3_P15 and P5_P15; *Timp1* at P5_P15; Figure 9d,i,j). A drastically increased expression compared to a down-regulation after hyperoxia directly after termination at P3 and P5 was observed for some transcripts after recovery (*Acta2*, *Col1a1*, *Col3a1* at P3_P15 and P5_P15; *Fn1* at P3_P15; Figure 9a–c,h). A direct hyperoxia-induced increase in expression was seen at P3 and P5 for *Egr1* (Figure 9g) and *tPa* (Figure 9l), for *CTGF* at P5 (Figure 9e), as well as for *Timp1* and *Timp2* at P3 (Figure 9j,k), some of which remained elevated without further oxygen-induced noxae (*CTGF* at P5_P15, *Edn1* at P3_P15, *Egr1* at P3_P15, *tPa* at P3_P15 and P5_P15). Caffeine affected the transcription of all target genes when administered in parallel with 80% oxygen exposure and, with few exceptions, caused a significant counter-regulation (Figure 9a–l).

Caffeine under control conditions modulated, except *Col1a1* and *tPa*, all investigated factors, predominantly directly after the end of caffeine administration (*Acta2*, *Col1a1*, *CTGF*, *Edn1* at P3 and P5; *Fn1*, *Timp2* at P3; *Egr1*, *Pu.1* at P5).

## 4. Discussion

Exposure of neonatal rats to 80% oxygen during the early postnatal period resulted in marked alterations in lung development, architecture, and molecular pathways regulating alveolarization, angiogenesis, and fibrosis. Although survival and body weight gain were not significantly affected, hyperoxia induced classical BPD-like pathology characterized by simplified alveolar structure, enlarged distal airspaces, reduced septal volume, and increased alveolo-capillary permeability. Morphological and molecular impairments were examined both during the acute phase of oxygen exposure, corresponding to the saccular stage and the transition to the alveolar stage, and later in adolescent rats during the alveolar stage without further insults, allowing assessment of immediate and longer-term effects on lung maturation. These morphological impairments were accompanied by downregulation of key alveolo-angiogenic genes, alongside transient activation of fibroblast-associated mediators. Notably, the injury progressed in a two-phase manner, comprising an acute, reversible barrier disruption with edema followed by chronic fibrotic remodeling that persisted through P15, which corresponds to the infant developmental stage and falls within the early alveolar stage of lung development.

Caffeine administration during hyperoxia markedly mitigated these structural and transcriptional abnormalities, preserving lung volume and architecture and largely restoring physiological gene expression profiles. Importantly, caffeine also exerted intrinsic developmental effects under normoxia by enhancing mediators of alveolarization and angiogenesis and suppressing pro-fibrotic and tissue-remodeling signals. Thus, while caffeine counteracted hyperoxia-induced injury, it also displayed context-dependent regulatory effects on postnatal lung maturation; anti-fibrotic under oxidative stress, yet transiently pro-fibrotic under normoxic conditions.

Overall, this study provides comprehensive evidence that caffeine preserves lung structure and function under oxidative stress while actively shaping physiological lung maturation, extending its known antioxidant and anti-inflammatory properties and suggesting therapeutic benefit for preventing or ameliorating neonatal lung injury. Compared with previous preclinical studies investigating the effects of caffeine in neonatal hyperoxia [46,53,54,70,71,72,73], the present work offers several unique contributions, driven by an enhanced methodological rigor. Earlier studies often lacked systematic morphometric lung endpoints, stereological quantification, or detailed phase-specific analyses. Our study overcomes these limitations by integrating high-resolution stereological and transcriptional analyses across two distinct postnatal exposure phases, namely the saccular and the alveolar stages. This approach allowed us to systematically evaluate organ-level remodeling, capture the stage-specific impact of hyperoxia and caffeine, and define the context-dependent, pro- versus anti-fibrotic effects of caffeine over time, which were previously unresolved. This combination of temporal resolution and morphometric rigor establishes a novel framework with high translational relevance to preterm infants.

The postnatal lung is highly vulnerable to oxidative injury given immature antioxidant defenses and ongoing alveolar and vascular growth. Hyperoxia downregulated essential alveolo-angiogenic mediators such as cyclin D2, FGF10, PDGFα/β, VEGF, and Tie2, while also dysregulating fibrotic mediators including Col1a1, CTGF, and TIMP1/2 [42,74]. Downregulation of Acta2/α-SMA must be interpreted within neonatal pathobiology: unlike adult fibrosis, neonatal hyperoxia primarily disrupts fibroblast subtype dynamics and alveolar septation rather than inducing classical collagen-rich fibrosis [75]. Lipofibroblasts and matrix fibroblasts, critical for septation, are selectively reduced under hyperoxia [76,77], explaining the apparent reduction in Acta2 and pro-fibrotic markers as a loss of myofibroblast commitment and impaired septation rather than fibrosis resolution [75,78].

Hyperoxia produced the characteristic simplified alveolar architecture repeatedly described in neonatal models, including enlarged distal airspaces, reduced septal volume and surface area, and increased permeability [42,74,79,80]. Persistent impairment of gas-exchange surface area is supported by transcriptional evidence of disrupted AT1/AT2 cell dynamics [81], consistent with increased MLI at later ages (P60) [81,82] and the structural deficits observed here up to the alveolar phase in rat pups (P15). The absolute surface area and mean thickness of the alveolar septa are key determinants of gas-exchange capacity; destruction of septa reduces total surface area and diffusion capacity, ultimately limiting oxygen uptake [83,84]. Early hyperoxia induced acute barrier disruption with permeability increases and interstitial edema without changes in total lung volume [85]; edema resolved after return to normoxia [86], but structural remodeling, increased parenchymal volume, and collagen deposition indicated progression toward chronic, likely irreversible injury. This two-phase pathophysiology, an acute, reversible edema followed by chronic structural reorganization, fits the persistent simplification observed after neonatal hyperoxia.

Neonatal programming effects can have lifelong implications. Neonatal hyperoxia causes long-term depletion of an AT2 precursor subset, which potentially impairs adult lung regeneration [81]. This aligns with epidemiological evidence showing increased risk of asthma, obstructive disease, and respiratory infections among individuals born preterm, independent of atopy, with risk increasing as gestational age decreases [11,87,88,89,90,91,92,93,94]. Datta et al. [95] confirmed that early, but not late, postnatal hyperoxia disrupts alveolarization and pulmonary arterial remodeling, and that mitochondria-targeted antioxidants protect alveolar development, paralleling the protective effects of caffeine in our model.

Hyperoxia triggers excessive ROS production, causing inflammation, apoptosis, mitochondrial dysfunction, and altered ECM turnover [42,50,51,79,95,96,97,98,99]. These effects impair lung development and recapitulate central features of BPD. Consistent with extensive prior work from our group using the same experimental cohort, caffeine reduced ROS accumulation, apoptosis, inflammatory signaling, and immune cell recruitment, while modulating cell cycle and ECM dynamics [50,51,100]. Hyperoxia-induced inflammatory responses involve macrophages with mixed activation states, neutrophils, and epithelial and endothelial cell subsets [51,79], all of which contribute to the injury cascade.

Vascular development is integral to alveolar growth and BPD pathogenesis. Hyperoxia strongly downregulated *FGF10*, VEGF, PDGF, and Tie2, all critical for vascular stability and microvascular maturation [39,101,102,103]. Experimental and human BPD consistently show disrupted VEGF signaling. Recent studies highlight the complex, dynamic regulation of VEGF in BPD and its potential as a biomarker [104]. While preclinical models suggest therapeutic benefits of VEGF modulation, safety concerns limit direct application [105]. Hyperoxia acutely suppressed PDGFα and its receptors, with post-recovery PDGFRα/β overexpression, and disrupted Tie2 signaling, indicating impaired mesenchymal diversity and vascular stability in our study. FGF10 and VEGF were also reduced, compromising microvascularization and contributing to BPD-like pathology [25,33,106].

As shown in this study, caffeine preserves lung architecture, normalizes lung volume, and counteracts transcriptional disturbances induced by oxygen toxicity, supporting its therapeutic potential. Its protective profile combines anti-inflammatory, antioxidant, anti-fibrotic, and antiapoptotic effects, with additional regulation of angiogenesis [46,49,50,51,107]. These mechanisms, including antioxidant and antiapoptotic actions, are consistent with findings from our previous studies in the same experimental cohort [49,50,51]. Caffeine reduces proinflammatory cytokine release, inhibits NLRP3 inflammasome activation, suppresses NF-κB signaling, and antagonizes adenosine A2A receptors, thereby modulating apoptosis and vascular development [46,51,108,109,110]. It enhances antioxidant enzyme expression [31,50], regulates pro-fibrotic mediators such as CTGF [111], and may improve pulmonary vascular remodeling [53,54]. Effects are concentration-dependent and can be paradoxical under certain conditions [46,52,55]. Beyond oxidative stress, caffeine modulates lung maturation under normoxia by upregulating FGF10, PDGFα, angiopoietin 1, and VEGFR2, while suppressing remodeling- and fibrosis-associated genes such as CTGF and Timp2, demonstrating context-dependent, multimechanistic regulation of postnatal lung development. Dumpa et al. [54] demonstrated that caffeine stabilizes HIF-2α, modulates VEGFR1, and improves lung and microvascular architecture, supporting its multifaceted role in developmental signaling.

Given this impact on angiogenic signaling, it is important to consider how alterations in the angiopoietin/Tie2/VEGF axis may affect vascular integrity in hyperoxia-induced lung injury. Dysregulation of this axis has been consistently linked to severe impairment of pulmonary vasculature, even when PECAM-1/CD31 levels remain stable [112]. Other hyperoxia models report concurrent reductions in PECAM-1 alongside changes in these mediators [113], and in chronic lung injury, PECAM-1 downregulation is associated with endothelial barrier loss and vascular remodeling [114]. Although our study did not directly assess PECAM-1/CD31, changes in Angpt1, Angpt2, Tie2, VEGF, VEGFR1, and VEGFR2 provide meaningful insight into microvessel density and endothelial function. Interpreting our molecular findings in the context of established PECAM/CD31 variability [112,115,116,117,118] acknowledges this limitation while underscoring the relevance of our approach for evaluating pulmonary microvascular integrity.

Caffeine significantly mitigates hyperoxia-induced lung injury. Hyperoxia increases ROS, DNA, and lipid damage and activates endogenous antioxidant pathways, including Nrf2/Keap1, superoxide dismutase, and heme oxygenase-1 [47,48,50,51,52,55]. Consistent with previous findings in the identical experimental cohort of newborn rat pups, our data demonstrate that caffeine reduces oxidative stress in neonatal lungs, reflected by decreased DNA damage, lower hydrogen peroxide levels, and modulation of antioxidant enzyme expression [50]. Mechanistically, this involves antagonism of adenosine A2A receptors, inhibition of cAMP/PKA-MAPK signaling, and attenuation of ER stress, thereby reducing alveolar apoptosis [47,48]. Consequently, caffeine acts as a potent antioxidant at clinically relevant doses.

Beyond oxidative stress, hyperoxia promotes fibrosis in the developing lung, particularly in preterm infants, via ROS-driven inflammation, TGF-β signaling, fibroblast–myofibroblast differentiation, and ECM accumulation, contributing to BPD pathogenesis [31,33,42,119,120,121,122,123,124,125,126,127]. Key profibrotic mediators, including Acta2/α-SMA, Col1a1, Col3a1, Fn1, Timp1/2, CTGF, and Gremlin-1, were induced by hyperoxia, with Gremlin-1 at P15 differentially modulated by caffeine in our study [128,129,130,131,132,133,134,135,136]. Caffeine exhibits antifibrotic effects by inhibiting TGF-β/CTGF signaling, reducing myofibroblast differentiation, ECM deposition, and airway remodeling [47,111,137,138,139,140,141,142]. Its mechanisms include blockade of TGF-β signaling, Nrf2 activation, and Snail1 suppression, with limited impact on Timp1/2, and modulation of MMP activity [99,141,142,143].

Caffeine’s regulation of fibrosis is context- and time-dependent. Under normoxia, we observed transient upregulation of TGF-β at P3 and CTGF at P5, indicating potential early pro-fibrotic signaling, whereas under hyperoxic stress, antioxidant and antifibrotic effects predominate. This reflects the complex, multi-stage process of pulmonary fibrosis, in which TGF-β drives myofibroblast differentiation and ECM accumulation [111,138,144]. Caffeine also modulates Smad signaling, reducing CTGF expression and downstream ECM production, thereby attenuating fibrotic remodeling. Furthermore, caffeine directly affects fibroblasts by influencing Acta2/α-SMA and other ECM-associated gene expression, limiting myofibroblast activation in injured lungs.

These effects are particularly relevant in the immature lung. In early postnatal development, excessive TGF-β and CTGF promote pathological matrix deposition, impaired alveolarization, and vascular malformation. In this context, caffeine can antagonize profibrotic signaling under oxidative stress, supporting structural lung maturation. Overall, our findings indicate that caffeine’s impact on the TGF-β/CTGF axis is strongly dependent on oxygen status and timing: under normoxia, transient pro-fibrotic effects may occur, while under hyperoxic stress, antioxidant and antifibrotic effects predominate, highlighting its dual, context-dependent role in postnatal lung development.

Caffeine therapy in preterm infants is well established for apnea of prematurity and is consistently associated with reduced BPD risk and improved respiratory outcomes [15,145,146,147,148]. Early initiation within the first two days of life further decreases BPD, mortality, and need for prolonged respiratory support [149,150,151]. Benefits may partly reflect reduced duration of mechanical ventilation and lower cumulative oxygen exposure [107,149,152]. Higher-dose regimens may provide additional pulmonary advantages, although long-term safety in extremely preterm infants requires caution [15,146].

Although caffeine citrate is an established and effective drug for the treatment of AOP, its dosage and application in clinical practice are still discussed, and further research is needed to determine the optimal doses and timing for its multiple beneficial effects on various organs [60]. It is important to note that while the benefits of caffeine are well documented in animal models, its effects outside the context of oxidative stress may have undesirable consequences, so its use beyond the recommended indications must be carefully considered [52].

## 5. Conclusions

Neonatal hyperoxia induced a biphasic, BPD-like lung injury, with an early, reversible barrier disruption followed by sustained fibrotic remodeling into adolescence, accompanied by impaired alveolo-angiogenic signaling and transient fibroblast activation. Caffeine counteracted these acute and long-term effects by preserving alveolar structure, maintaining angiogenic gene expression, and limiting fibrosis via TGF-β/CTGF modulation. Its effects were context-dependent, being largely protective under hyperoxia but briefly pro-fibrotic under normoxia. These findings highlight caffeine’s capacity to modulate postnatal lung maturation and its therapeutic promise for preterm infants at risk of BPD.

## Figures and Tables

**Figure 1 antioxidants-14-01497-f001:**
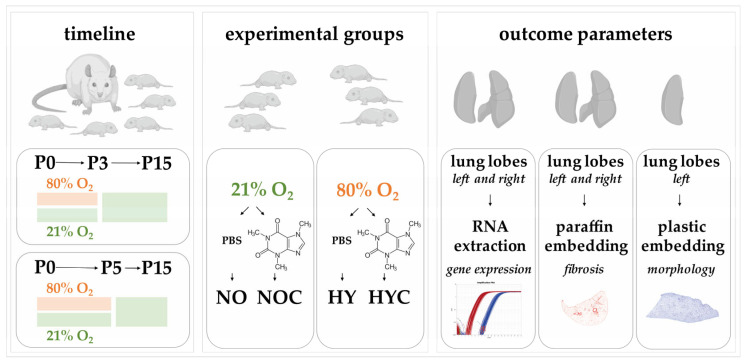
Schematic of the experimental design showing oxygen exposure (normoxia or hyperoxia, **left box**) with or without caffeine treatment from P0 to P3 or P5, followed by analyses either immediately (P3, P5) or after recovery in room air (P15). Four groups were studied at each time point: normoxia with vehicle (NO), normoxia with caffeine (NOC), hyperoxia with vehicle (HY), and hyperoxia with caffeine (HYC, **middle box**). Outcome parameters (**right box**) included RNA gene expression, Sirius Red staining (paraffin sections), and morphology (plastic embedding).

**Figure 2 antioxidants-14-01497-f002:**
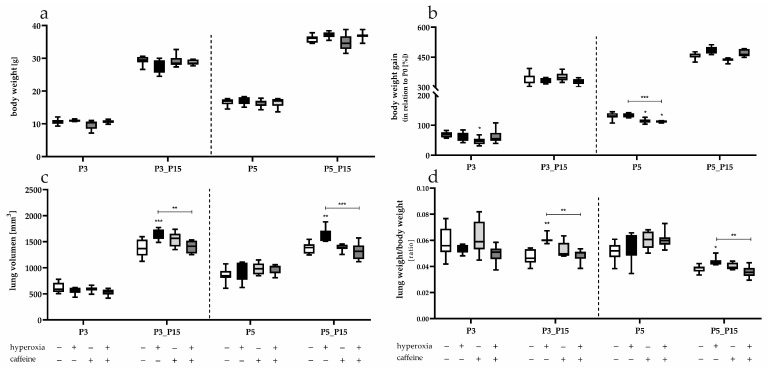
Quantification of (**a**) body weight on examination day, (**b**) body weight gain over time, (**c**) lung volume on examination day, and (**d**) the ratio of lung weight to body weight for 3 days postnatal oxygen exposure (P3) and recovery (P3_P15) as well as 5 days postnatal oxygen exposure (P5) and recovery (P5_P15). Data are presented as median of the experimental groups (normoxia, white; hyperoxia, black; normoxia with caffeine, light gray; hyperoxia with caffeine, dark gray) or normalized to the level of rat pups exposed to normoxia at each time point (body weight gain, control 100%, white bars). *n* = 7–8 per group; * *p* < 0.05, ** *p* < 0.01, *** *p* < 0.001 (ANOVA; Kruskal–Wallis; Brown–Forsythe).

**Figure 3 antioxidants-14-01497-f003:**
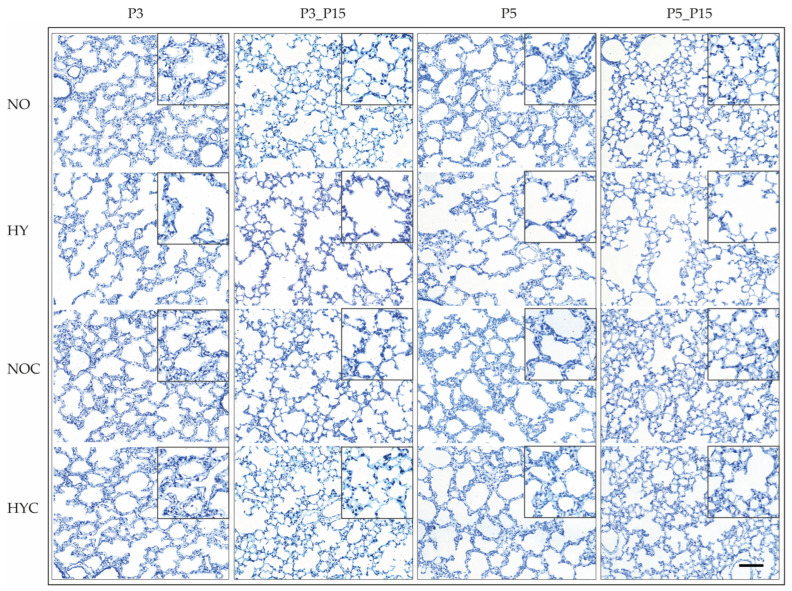
Representative toluidine stain imaging of left lung lobes for 3 days postnatal oxygen exposure (P3, **left**) and recovery (P3_P15, **middle left**) as well as 5 days postnatal oxygen exposure (P5, **middle right**) and recovery (P5_P15, **right**) of rats exposed to normoxia (NO) or hyperoxia (HY) compared to lung lobes of rats treated with caffeine (NOC, HYC). Scale bar 100 µm. The outlined regions (black bordered boxes) are magnified 1.8× compared to the original image.

**Figure 4 antioxidants-14-01497-f004:**
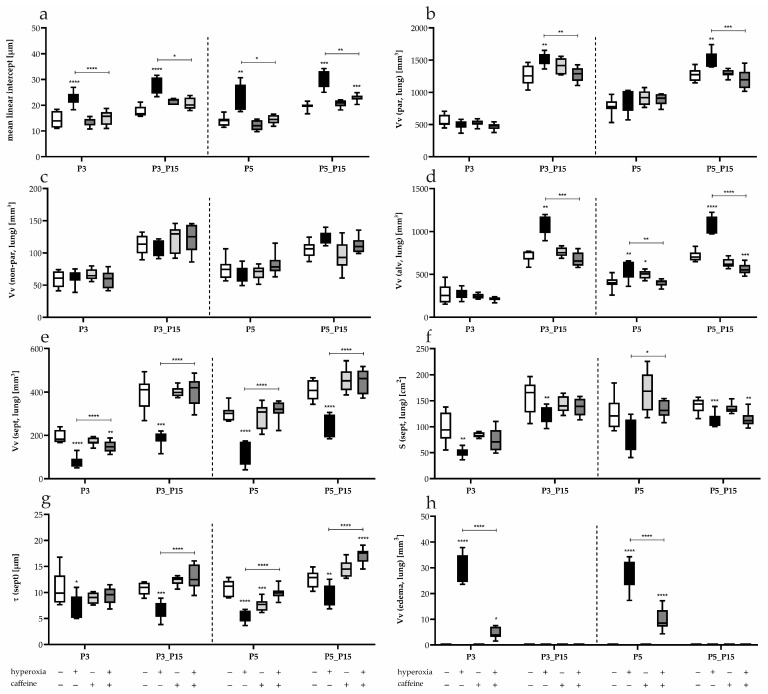
Quantification of (**a**) mean linear intercept (MLI), volume of (**b**) parenchyma (V_V_ (par, lung)), (**c**) non-parenchyma (V_V_ (non-par, lung)), (**d**) alveolar airspace (V_V_ (alv, lung)), (**e**) alveolar septa (V_V_ (sept, lung)), (**f**) absolute surface of the alveolar septum (S (sept, lung)), (**g**) mean thickness of the alveolar septum (τ (sept)), and (**h**) alveolar edema (V_V_ (edema, lung)) for 3 days postnatal oxygen exposure (P3) and recovery (P3_P15) as well as 5 days postnatal oxygen exposure (P5) and recovery (P5_P15). Data are presented as median of the experimental groups (normoxia, white; hyperoxia, black; normoxia with caffeine, light gray; hyperoxia with caffeine, dark gray). *n* = 6–8 per group; * *p* < 0.05, ** *p* < 0.01, *** *p* < 0.001, **** *p* < 0.0001 (ANOVA; Kruskal–Wallis; Brown–Forsythe).

**Figure 5 antioxidants-14-01497-f005:**
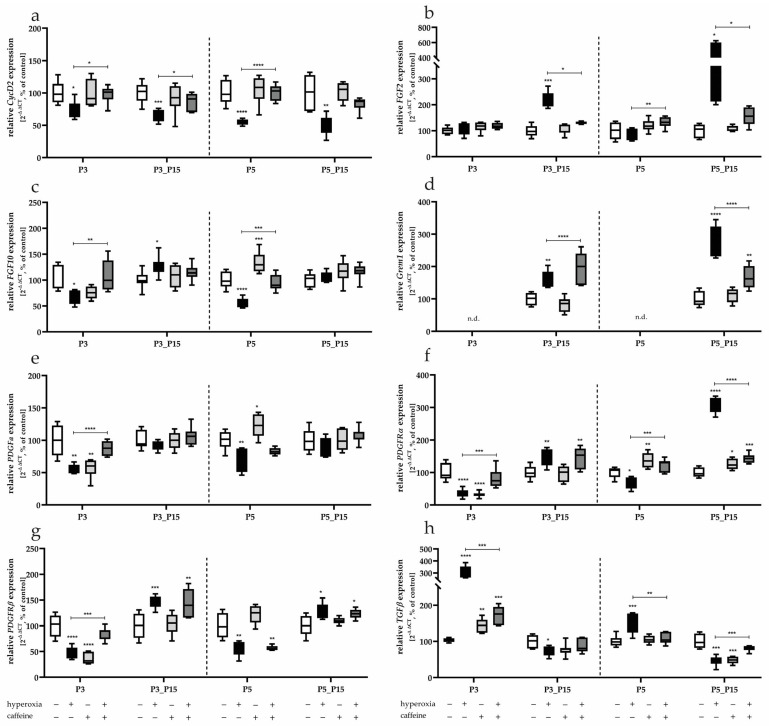
Quantification of alveolarization mediator’s transcript expression of a whole lung lobe with (**a**) *CycD2*, (**b**) *FGF2*, (**c**) *FGF10*, (**d**) *Grem1*, (**e**) *PDGFα*, (**f**) *PDGFRα*, (**g**) *PDGFRβ*, and (**h**) *TGFβ* for 3 days postnatal oxygen exposure (P3) and recovery (P3_P15) as well as 5 days postnatal oxygen exposure (P5) and recovery (P5_P15). Data are normalized to the level of rat pups exposed to normoxia at each time point (control 100%, white bars) with verum groups hyperoxia (black), normoxia with caffeine (light gray), and hyperoxia with caffeine (dark gray). *n* = 7–8 per group; * *p* < 0.05, ** *p* < 0.01, *** *p* < 0.001, **** *p* < 0.0001 (ANOVA; Kruskal-Wallis; Brown–Forsythe).

**Figure 6 antioxidants-14-01497-f006:**
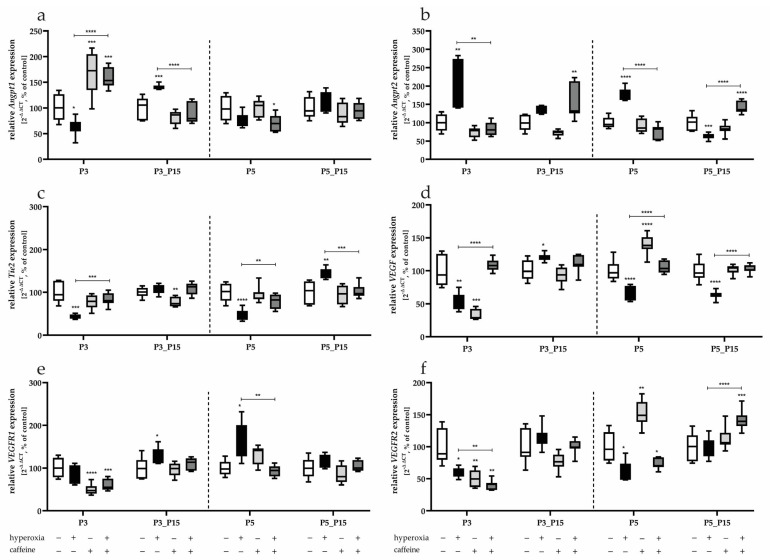
Quantification of angiogenesis mediator’s transcript expression of a whole lung lobe with (**a**) *Angpt1*, (**b**) *Angpt2*, (**c**) *Tie2*, (**d**) *VEGF*, (**e**) *VEGFR1*, and (**f**) *VEGFR2* for 3 days postnatal oxygen exposure (P3) and recovery (P3_P15) as well as 5 days postnatal oxygen exposure (P5) and recovery (P5_P15). Data are normalized to the level of rat pups exposed to normoxia at each time point (control 100%, white bars) with verum groups hyperoxia (black), normoxia with caffeine (light gray), and hyperoxia with caffeine (dark gray). *n* = 7–8 per group; * *p* < 0.05, ** *p* < 0.01, *** *p* < 0.001, **** *p* < 0.0001 (ANOKruskal–Wallisllis; Brown–Forsythe).

**Figure 7 antioxidants-14-01497-f007:**
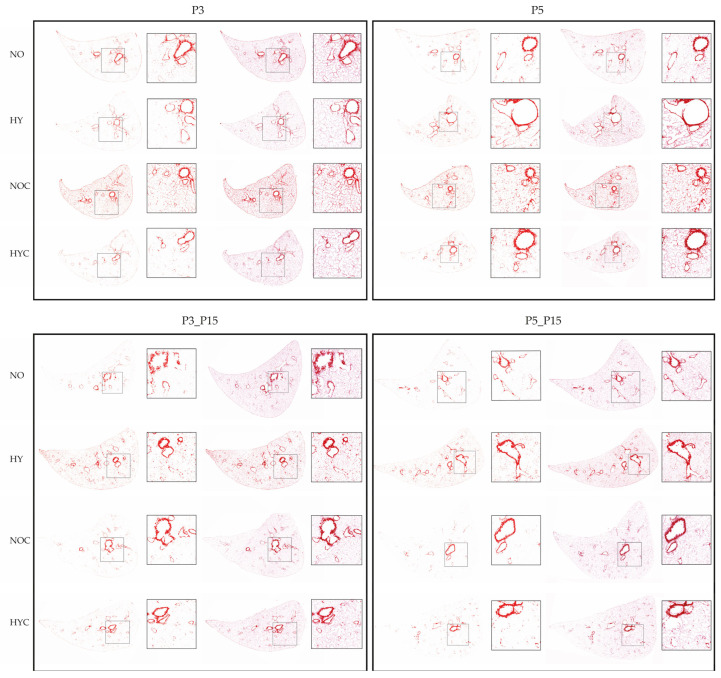
Sirius red stain imaging representative of general collagen for 3 days postnatal oxygen exposure (P3, **upper left**) and recovery (P3_P15, **bottom left**) as well as 5 days postnatal oxygen exposure (P5, **upper right**) and recovery (P5_P15, **bottom right**) in lung lobes of rats exposed to normoxia (NO) or hyperoxia (HY) compared to lung lobes of rats treated with caffeine (NOC, HYC). Images indicated stained Sirius Red areas (threshold, **left**) and merged with brightfield images (**right**).

**Figure 8 antioxidants-14-01497-f008:**
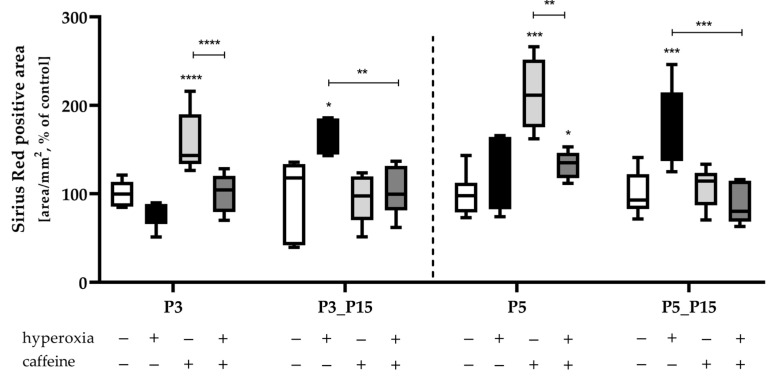
Quantification of Sirius Red positive areas for 3 days postnatal oxygen exposure (P3) and recovery (P3_P15) as well as 5 days postnatal oxygen exposure (P5) and recovery (P5_P15). Data are normalized to the level of rat pups exposed to normoxia at each time point (control 100%, white bars) and represented hyperoxia (black), normoxia with caffeine (light gray), and hyperoxia with caffeine (dark grey). *n* = 6–8 per group; * *p* < 0.05, ** *p* < 0.01, *** *p* < 0.001, **** *p* < 0.0001 (ANOVA; Kruskal–Wallis; Brown–Forsythe).

**Figure 9 antioxidants-14-01497-f009:**
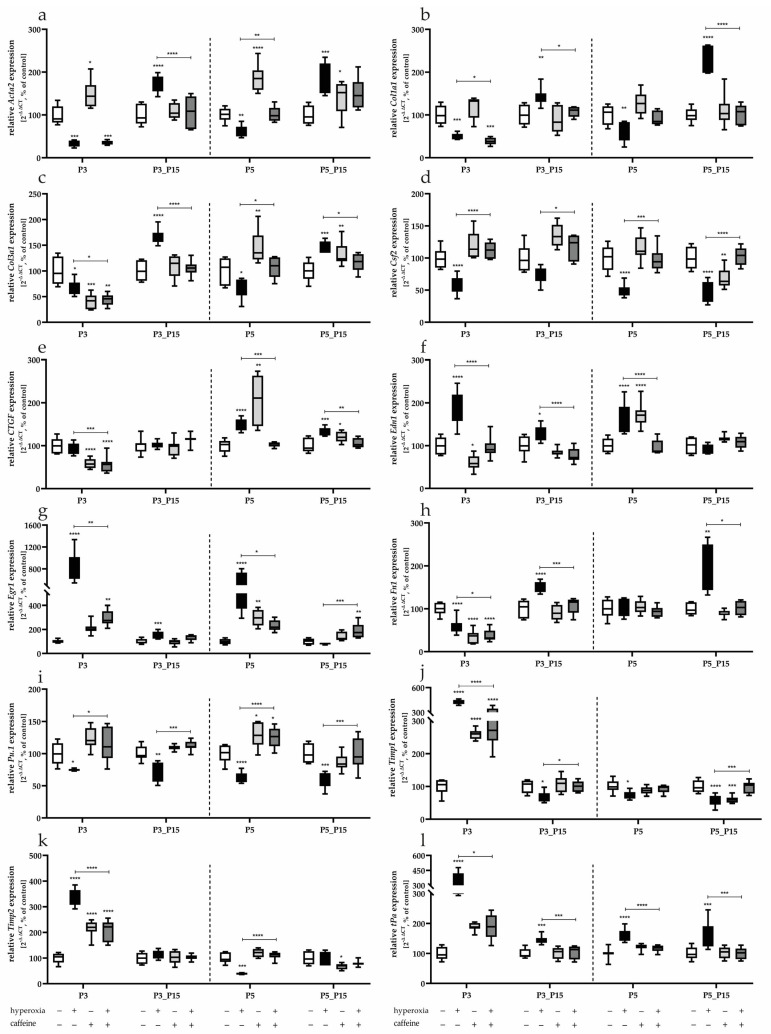
Quantification of fibroblast-associated mediator’s transcript expression of a whole lung lobe with (**a**) *Acta2*, (**b**) *Col1a1*, (**c**) *Col3a1*, (**d**) *Csf2*, (**e**) *CTGF*, (**f**) *Edn1*, (**g**) *Egr1*, (**h**) *Fn1*, (**i**) *Pu.1*, (**j**) *Timp1*, (**k**) *Timp2*, and (**l**) *tPa* for 3 days postnatal oxygen exposure (P3) and recovery (P3_P15) as well as 5 days postnatal oxygen exposure (P5) and recovery (P5_P15). Data are normalized to the level of rat pups exposed to normoxia at each time point (control 100%, white bars) with verum groups hyperoxia (black), normoxia with caffeine (light gray), and hyperoxia with caffeine (dark grey). *n* = 7–8 per group; * *p* < 0.05, ** *p* < 0.01, *** *p* < 0.001, **** *p* < 0.0001 (ANOVA; Kruskal–Wallis; Brown–Forsythe).

## Data Availability

The data used to support the findings of this study are available from the corresponding author upon request. The oligonucleotides used for qPCR and their target gene sequences are summarized in the Appendix A.

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
