# Peer review of "Caffeine Protects Against Hyperoxia-Induced Structural Lung Injury and Restores Alveolar Development in Neonatal Rats"

_antioxidants, 2025, doi:10.3390/antiox14121497_

Round 1

Reviewer 1 Report

I have carefully reviewed the article entitled “Caffeine Prevents Oxygen-Induced Structural Lung Damage of Newborn Rats but Induces Profibrotic Remodeling in Normoxic Controls”. The work addresses an important clinical and translational question concerning caffeine’s dual effects on hyperoxia-induced lung injury and fibrosis in neonatal models. However, some minor concerns must be addressed to enhance the scientific clarity and presentation of the manuscript. Below are section-wise detailed comments.

Comments

Abstract

  • Please follow the abstract structure 1. Background, 2. Aim, 3. Methodology, 4. Results and 5. Conclusion.
  • Include a concise summary of the main findings and potential future directions at the end of the abstract.

Introduction

  • L88-107: Combine overlapping statements on ROS and immature antioxidant defenses.
  • L96-100: Discuss any controversies or recent updates regarding VEGF signaling in BPD to strengthen novelty.
  • L128-132: Highlight clearly what this study adds compared to prior caffeine studies; mechanistic, morphological, or gene-expression dimension.
  • Consider elaborating more on the research background, clearly stating the knowledge gap, the problem being addressed, and how the present study aims to provide the solution in the introduction section.
  • Conclude the introduction with a clear hypothesis and concise study aim statement.

Materials and Methods

  • L150-175: Include a justification for caffeine dose selection (10 mg/kg) with reference.
  • L319-330: Clarify which post-hoc test was used.

Discussion

  • The discussion is too lengthy; condense to focus on the most important mechanisms and comparisons with previous studies.
  • L592-597: Add a brief mechanistic link between antioxidant properties of caffeine and the observed reduction in fibrotic gene expression.
  • L633-652: The section well describes caffeine’s mechanisms; however, the paragraph could be split to enhance readability.

Conclusion

  • Conclusion section should be in 3-4 concise sentences emphasizing key findings, significance, limitations, and future prospects.

As above

Reviewer 2 Report

This study outlines the effects of caffeine administration on alveolar simplification, angiogenesis and fibrosis in newborn rat pups exposed to 3 or 5 days of hyperoxia from day of birth, with immediate assessment as well as after recovery in room air at post-natal day 15. The authors have used histopathological and volumetric analyses of the lungs, as well as gene expression to examine molecular pathways in neonatal lung development that are disrupted by hyperoxia and in turn how they are modulated by caffeine administration. As the authors state, caffeine is an antioxidant, and with oxidative stress being a key pathogenic component of hyperoxia-induced lung injury, it is a notable omission that oxidative stress pathways have not been assessed in this study. There is also no assessment of inflammation, either in the form of inflammatory cytokines or cellular analysis/staining. The findings of this study would be strengthened significantly by additional analyses if these are possible, and at the very least should be discussed as a limitation. The title suggests profibrotic remodelling in normoxic controls but this should be toned down in light of methodology question below. The Discussion is extensive and extremely long, with repetition in many areas that could be removed to improve relevance and clarity.

Specific comments are listed below.

  1. Methods 2.2: It is not clear whether caffeine was administered for the entire duration of the study for the pups recovering in room air following hyperoxia exposure or only during the PN0-PN5 period. Could this please be clarified by the authors? i.e. PN3 group received one dose, PN5 group received 2 doses. This has implications for the acute effects of caffeine observed at the PN3 and PN5 groups if the PN3-15 and PN5-15 did not receive caffeine following this point. This should also be referred to when discussing the results.
  2. Results Figure 2d and corresponding text and Figure legend (Page 8): This should be lung volume? It does not appear that lung weights were measured, which would have given an indication of pulmonary edema instead. Please correct.
  3. Results Figure 4 page 10: Could alveolar septal wall thickness also be measured on the existing tissue sections? This will provide additional insight into the alveolar effects of caffeine administration.
  4. Results Figure 6 Page 12: Did the authors also perform gene expression for Pecam-1/CD31? Vascular regression is a key component of disrupted angiogenesis and it would be valuable to see this evaluated directly in addition to the angiogenesis mediators.
  5. Results Figure 8 page 14: This result in particular highlights point 1 above: Caffeine increased collagen content at PN3 and PN5 in normoxia but it is not clear if the P3_P15 and P5_15 groups also continued to receive caffeine until their endpoint or if it was only until P3 and P5 respectively. Please clarify.
  6. Results 3.3.2 Fibrosis-associated mediators: Could the authors please provide a balanced discussion of fibroblast subtypes and their importance in alveolarization in the developing lung? The designation of these genes as only fibrosis-inducing is a bit misleading, since specific fibroblast subpopulations are selectively depleted by hyperoxia. Acta2 is also expressed by smooth muscle cells, and as part of the vascular regression phenotype inherent in hyperoxia this is expected to be reduced, however it does not fit with a pro-fibrotic phenotype if the discussion is solely centred on fibroblasts.
  7. Discussion paragraph 1 line 526-527: As above, outline fibroblasts as key cell types in alveolarization rather than just fibrosis-inducing mediators.
  8. Discussion paragraphs 4 and 5 (lines 542-552) are repetitive- points are mentioned in paragraph 1. Please remove or consolidate.
  9. Discussion lines 553-605: Very exploratory and could be summarized/removed as none of these points were assessed or alluded to in the current study. Line 606-632 can also be condensed significantly,
  10. Discussion lines 653-661: repetitive.
  11. The description of antioxidant effects of caffeine in lines 662-680 highlights that this was not done in the current study- condense or state with relation to current study.
  12. Lines 686-770: must be condensed or removed, this is too descriptive and loses relevance to the current study.

General comments:

  1. Oxidative stress and Inflammation were not formally assessed, so these should be discussed as a limitation to the current study.

Reviewer 3 Report

The manuscript by Endesfelder, et al entitled “Caffeine prevents oxygen-induced structural lung damage of newborn rats but induces profibrotic remodeling jn normoxic controls” examines the effect of caffeine on lung structure in a model of bronchopulmoniary dysplasia (BPD) induced by exposure to high concentrations of oxygen (hyperoxia).  Bronchopulmonary dysplasia is a disease found in oremature infants and has significant effects on mortality and morbidity, particularly on lung function as a child or adult.  The authors do a good job on outlining the importance of improving the understanding of the underlying pathophysiology of BPD and the need to identify effective therapies.  In this manuscript the authors explore the effect of caffeine on regulating the effect of hyperoxia on oxidative stress and lung fibrosis.  Caffeine has been previously studied on its effects on BPD, thereby decreasing the novelty of the study.  The animal model used in the study is appropriate and is well described in the manuscript.  Treatment with caffeine was shown to decrease augmented lung volumes induced by hyperoxia and improve alveolization.  To examine mechanisms regulating these findings, the researchers examine lung RNA expression for genes regulating alveolization, angiogenesis, and fibrosis.  Overall this RNA data is difficult to follow as expression patterns vary depending on whether one looks at the acute hyperoxia exposure period (2 separate periods) and the recovery phase.  Differences are also present depending on treatment with caffeine.  These changes are not consistent across conditions and therefore increases the complexity in interpretation.  The manuscript would be improved by focusing on one area of gene expression with subsequent mechanistic studies to further explore that pathway.  At present the manuscript is overall descriptive.   A defined pathway analysis would also strengthen the manuscript.  The effects of caffeine on regulating lung fibrosis, being both pro- and anti-fibrotic depending on experimental condition, are interesting but makes it unclear if caffeine has a potential role in the treatment of BPD.  The Discussion section can be significantly shortened.

See above

Round 2

Reviewer 2 Report

Thank you for revising the manuscript and addressing all concerns.

No further comments.

Reviewer 3 Report

The revised manuscript is significantly improved.  There rewritten Discussion section now placed all the research findings into context and enhance the impact of the findings.  

Please see above.